# Sire Effects on Post-Weaning Growth of Beef-Cross-Dairy Cattle: A Case Study in New Zealand

**DOI:** 10.3390/ani10122313

**Published:** 2020-12-07

**Authors:** Natalia Martín, Nicola Schreurs, Stephen Morris, Nicolás López-Villalobos, Julie McDade, Rebecca Hickson

**Affiliations:** 1School of Agriculture and Environment, Massey University, Private Bag 11 222, Palmerston North 4442, New Zealand; N.M.Schreurs@massey.ac.nz (N.S.); S.T.Morris@massey.ac.nz (S.M.); N.Lopez-Villalobos@massey.ac.nz (N.L.-V.); R.Hickson@massey.ac.nz (R.H.); 2Greenlea Premier Meats Ltd., P.O. Box 87, Hamilton 3240, New Zealand; Julie@greenlea.co.nz

**Keywords:** sire, genetics, dairy-beef, beef-on-dairy, live weight, crossbreeding, growth rate

## Abstract

**Simple Summary:**

Cattle born in the dairy industry are a very important source of beef. This study evaluated the growth of calves born to dairy cows and sired by a range of Angus and Hereford bulls. The sire had effects on live weights of their progeny at all ages. Growth trajectories differed among beef sires and the differences in live weights between the lightest and heaviest sires increased as live weight increased. Live weight at early age could explain a large proportion of live weights at later ages. Thus, using beef sires selected for growth has the potential to increase the live weight of cattle born on dairy farms for meat production.

**Abstract:**

Little is known about the growth performance of beef sires used over dairy cows in New Zealand. This experiment aimed to evaluate the growth of Angus and Hereford sires via progeny testing of beef-cross-dairy offspring born to dairy cows and grown on hill country pasture. Live weights at 131, 200, 400, 600 and 800 days were analysed from a dataset of 5208 records from 1101 progeny of 73 sires. The means of the progeny group means for live weight were 118.6 kg at 131 days, 159.1 kg at 200 days, 284.2 kg at 400 days, 427.0 kg at 600 days and 503.6 kg at 800 days, and the overall daily growth rate was 0.58 kg/day from 131 to 800 days. The sire affected (*p* < 0.05) the live weight of their progeny at all ages. Differences in live weights between the lightest and heaviest progeny group means increased from 19 kg at 131 days to 90 kg at 800 days. Even though growth of calves was likely restricted to 200 days, live weight at 200 days explained 51–56% of the variation in live weights at 400 and 600 days (*p* < 0.05). Thus, the use of beef sires selected for growth has the potential to increase the live weight of cattle born on dairy farms for meat production.

## 1. Introduction

Cattle born in the dairy industry contribute around 66% of New Zealand’s beef production on a per head basis [1,2,3]. Most cattle destined for beef production are farmed on hill country pasture until slaughter with minimal supplementation [4]. Animals that are more efficient in converting pasture into body weight are more desirable in this type of system, because they grow faster and reach the target slaughter weight at a younger age, and because they require less feed to achieve same results compared with a less efficient animal [5,6]. Similarly, animals with the same mature size and maturation rate, but greater yearling growth rate [7], will be wanted for this system as they may need less time and feed to achieve their slaughter weight. Growth path can have important effects on carcass characteristics as well, most of which relate to the composition of the animal, particularly fatness [8]: cattle on a fast growth path generally have greater fat thickness and intramuscular fat than cattle that grow slowly. These differences in fat content are likely to be a result of both diet and growth rate. Thus, live weight and growth trajectories of each animal are important.

Crossbreeding is an effective tool for increasing performance and profitability through heterosis [9,10]. Crossbreeding using beef-breed sires over dairy cows can increase income from sales of the beef-cross-dairy calves born on the dairy farm. The increase in the price of these calves is based on the expected greater performance, that is, higher growth rates producing heavier animals with higher dressing out or meat yield percentages compared with dairy-bred cattle [11,12]. Furthermore, there is a possibility to identify sires with good genetic merits for both calving ease and carcass weight [13]. These sires will produce calves that have lighter birth weights but high growth rates, making them desirable for dairy-beef systems, as they will ensure calving success (particularly important in young cows at their first parturition) without compromising meat production. Given that the terminal sire has a considerable direct genetic effect on the growth potential of the progeny in crossbreeding systems, then beef sires with improved genetics can be used to generate beef-cross-dairy calves better able to satisfy the requirements of the dairy, grower and finisher farmers, through to the meat processor.

Dairy farmers use genetic information or the estimated breeding values (EBVs) of dairy sires calculated by New Zealand Animal Evaluation Limited [14], with the aim of obtaining progeny of higher genetic merit for farm profit than the average of the herd. However, dairy farmers that use beef sires on specific groups of cows (e.g., the late calving cows or primiparous heifers) do not necessarily acquire sires with genetic records. Angus and Hereford are breeds widely used in New Zealand farming systems [15], and both breeds have selection programs to improve the genetic merit for beef production. Coleman [16] demonstrated that sires of Angus and Hereford breeds have a wide variation in performance for gestation length and birthweight, two very important traits for the dairy farmer, and that these traits are well predicted by their EBVs. These beef sires also have a range of EBVs for growth or live weight later in life (200, 400 and 600 days of age). However, EBVs are calculated by BREEDPLAN [17] within each specific breed. Little is known about the performance of beef sires for growth traits in a dairy-beef system with beef-cross-dairy cattle. The hypothesis is that growth trajectories differ among beef sires used to produce beef-cross-dairy cattle that graze on hill country conditions. Beef sires may change ranking as the progeny grows and some sires will be considered “curve benders”, having light progeny at birth but heavy progeny at slaughter. 

The aim of this experiment was to evaluate live weights post-weaning, from day 131 to 800, of a selection of Angus and Hereford sires via progeny testing of beef-cross-dairy offspring grown on hill country pasture. 

## 2. Materials and Methods 

This experiment was conducted at Limestone Downs, near Port Waikato, New Zealand (37°28′ S, 174°45′ E) with approval from the Massey University Animal Ethics Committee (15/65 and 18/50).

### 2.1. Animals 

Angus-sired and Hereford-sired calves born to dairy cows in spring 2016 (*n* = 584) and 2017 (*n* = 517), were included in the project. Calves were born to 2-year-old (primiparous) and mixed-aged (3+ years old, multiparous) dairy cows, which were predominantly Holstein–Friesian or Holstein–Friesian–Jersey crossbred. Calves were 131.4 days old (SD 17.2, 4 months of age) and 124.8 kg (SD 16.0) at the start of the experiment. Heifers went to slaughter at a mean age of 823 days of age (range 693–934) targeting an average live weight of 500 kg, whereas steers went to slaughter at a mean age of 887 days of age (range 821–955) targeting an average live weight of 600 kg. From the initial 1101 calves, a total of 915 cattle reached 800 days of age (*n* = 429 born in 2016 and *n* = 486 born in 2017). Of the 186 cattle that did not reach 800 days, 111 were processed prior to 800 days and 75 were excluded from the project prior to slaughter as detailed in the data cleaning section.

#### 2.1.1. Beef Sire Selection

Lactating mixed-aged cows were individually inseminated with semen from Angus and Hereford bulls, as described by Coleman [16]. Semen was rotationally allocated to mating days, and randomly allocated to cows in oestrus on each mating day. Cows were bred in spring for 63 days in 2015 and 54 days in 2016. Sires were selected for the project based on their EBVs so that, within each breed, a spread of birth weight, gestation length and live weight at 600 days of age was achieved. Birth weight EBVs were restricted to the lighter 50% of the breed at the time of selection. When similar sires were available, those with superior EBVs for intramuscular fat and eye muscle areas were selected. The sire team over the 2 seasons included a total of 31 Angus and 34 Hereford bulls, of which 14 Angus and 11 Hereford sires were used in both seasons. 

The 15-month-old replacement dairy heifers were joined with either Angus or Hereford bulls by natural mating. Each breed was used in separate groups of heifers and heifers were randomly allocated to each group to ensure that a similar number of heifers were mated to each breed. Heifers were mated in spring for 69 days in 2015 with 4 bulls of each breed, and 66 days in 2016 with 2 bulls of each breed. Sires for the first calving heifers were selected to be in the lightest 15% of the breed for birth weight. The sire team over the 2 seasons included a total of 6 Angus and 6 Hereford bulls, all of which were used for one season only.

The EBV of each sire was obtained from the online databases of Angus and Hereford breed associations [17]. The data collected in this experiment were not included for the calculation of the BREEDPLAN EBVs for these sires. Mean and range of EBVs for weight traits by breed of sire are presented in Table 1. 

#### 2.1.2. Calving and Rearing at the Dairy Farm, before Current Experiment 

All calves had date of birth, sex and live weight recorded within 24 h of birth, and were DNA parentage verified as part of a previous experiment described by Coleman [16]. Mean birth date was 3 August 2016 and 5 August 2017 for the calves included in the present experiment, with an overall mean birth weight of 36.4 kg (SD 4.7). Fewer than 1% of calves born to mixed-aged cows and fewer than 7% of calves born to 2-year-old cows (8% in 2016 and 4% in 2017) were assisted at calving [16]. Calves were artificially reared on an allowance of 4 L of milk/head/day, and calf meal (crude protein (CP) 17 to 20% [16]) was offered during the transition from milk to pasture. Calves were weaned at a minimum of 85 kg live weight. Once weaned, calves were moved from the dairy platform to the sheep and beef hill country platform of the same farm. Male calves were castrated before 4 months of age.

### 2.2. Feed Management at the Beef Farm

At a mean age of 131.4 days (SD 17.2, 131 days), calves were allocated to 6 grazing herds based on live weight (light, intermediate and heavy) and sex (female and male) and balanced for sires so that, where possible, all sires were represented in each grazing herd within year. In total, there were 12 grazing herds (2 years × 2 sexes × 3 liveweight groups), and animals remained in those herds throughout the experiment.

All cattle were grazed on summer-dry hill country pasture on the coastal farm under commercial conditions. Typical pasture species at Limestone Downs farm included ryegrass (*Lolium perenne*), kikuyu (*Pennisetum clandestinum*), ratstail (*Sporobolus africanus*) and some legumes such as white clover (*Trifolium repens*) and birdsfoot trefoil (*Lotus corniculatus*), with varying compositions throughout the year [18]. Overall, pasture was of medium to low quality, with mean metabolizable energy (ME) 10.1 MJ (range 7.0–12.5) per kilogram of dry matter (DM)—crude protein (CP) 18.8% DM (range 5.1–31.5), neutral detergent fibre (NDF) 48.9% DM (range 32.7–63.3) and organic matter digestibility (OMD) 69.3% DM (range 50.9–84.0). Particularly low quality was found during summer (December to February), with high pasture covers (average 2968 kg DM per hectare) but low energy (9.2 MJME/kgDM), low protein (CP 12.6% DM) and high fibre content (NDF 55.8% DM).

Two herds of the 2016-born heifers were supplemented with 1 kg/head/day of calf meal (CP 15%, with the main ingredients being barley, maize and wheat), in March 2017 (intermediate for 37 days and heavy for 28 days). Two herds of the 2017-born steers were intermittently supplemented with palm kernel and tapioca at 2 kg/head/day during May–September 2019 (intermediate for 60 days and light for 83 days). No other supplementation was provided.

### 2.3. Live Weight Measurements

Animals were weighed on the farm through a weigh crate (cattle crush model Cattlemaster Titan, made by Te Pari Products Ltd., Oamaru, New Zealand; weight scales model XR5000, Tru-Test, Auckland, New Zealand) within 3 h after yarding from a nearby paddock. Live weights were recorded to the nearest 0.5 kg up to 200 kg live weight, and to the nearest 1.0 kg beyond 200 kg. Live weights were measured at monthly intervals from 4 to 12 months of age, and then at a minimum of 2-monthly intervals until slaughter (between 22.7 and 31.3 months of age).

### 2.4. Statistical Analysis

#### 2.4.1. Data Cleaning

Animals born to sires with a minimum of 5 progeny were included in analysis (excluded *n* = 9 progeny of 3 sires in 2016 and *n* = 3 progeny of 1 sire in 2017). Animals that went missing, were recorded to have ill health, were removed from their grazing herd for more than 2 months or that died were excluded from analysis of traits measured after they first left their herd (*n* = 47 in 2016 and *n* = 28 in 2017). The dataset consisted of 23,426 liveweight records from 1101 progeny of 73 sires.

#### 2.4.2. Predicting Live Weights at Specific Age

Short-term fluctuations in live weight were smoothed out by calculating centred moving averages of 3 liveweight records per animal (using previous, current and subsequent liveweight records) using the Expand procedure (SAS 9.4, SAS Institute Inc., Cary, NC, USA). Predicted live weights for each animal at 131, 200, 400, 600 and 800 days were calculated by interpolation of the smoothed liveweight curves. The dataset used for analysis consisted of 5208 predicted weights from 1101 progeny of 73 sires.

Predicted live weights (33,111 records) were fitted to actual live weight data (23,426 records) using a regression model. Predicted live weights were used to plot growth curves across age for all progeny, grouping by sex of the animal. Average daily live weight gain (in kilograms per day) was calculated as the difference of the predicted live weights divided by the number of days between two ages.

#### 2.4.3. Contemporary Groups

For comparisons among sires, the contemporary group was defined as the group of animals in the same grazing herd (*n* = 6) and year (*n* = 2, 2016 and 2017), that were progeny of dams of the same age (*n* = 2, 2-year-old and mixed-aged) and progeny of sires of the same breed (*n* = 2, Angus and Hereford). Sex was already accounted for in the grazing herd effect. These contemporary groups (*n* = 48) had between 4 and 47 animals, with an age range of 11 to 65 days between the youngest and oldest animal in each group. For analysis at 131 days, the contemporary groups included year, sex, age of dam and breed of sire (2 × 2 × 2 × 2 = 16 groups). One contemporary group (light Angus-sired steers born to 2-year-old cows in 2016) had 2 animals from 1 sire from 600 days, and so was excluded from analysis for 600 and 800 days.

#### 2.4.4. Statistical Models

Linear mixed models (SAS 9.4, SAS Institute Inc., Cary, NC, USA) were used to estimate least-squares means of the progeny groups for live weight at 131, 200, 400, 600 and 800 days. The model to compare sires included the fixed effect of sire within breed, and the random effect of the contemporary group (*n* = 48). An overall mean of the least-squares means of the progeny groups was calculated for each age, with equal weighting per sire regardless of number of progeny.

Least-squares means of the progeny groups for live weights at 400, 600 and 800 days were regressed against least-squares means of the progeny groups for live weights at 131 and 200 days using random regression weighted by the number of progeny for each sire.

## 3. Results

### 3.1. Growth Curves of Beef-Cross-Dairy Progeny

Growth curves with the predicted live weights calculated with centred moving averages of three values are presented in Figure 1. These predicted live weights were a close fit to actual live weights (*p* < 0.001, *R*^2^ = 99.75%).

### 3.2. Live Weights by Sire

The total number of progeny and the mean of the least-squares means of the progeny groups for live weights at 131, 200, 400, 600 and 800 days are presented in Table 2 and their distribution is graphed in Figure 2. The sire affected the live weight of their progeny at all ages (*p* < 0.05, Table 2). The breed of the sire had no effect on live weight at any age from 131 to 800 days (*p* > 0.05). Cattle grew at an overall rate of 0.58 kg/day, with a mean of 0.59 kg/day from 131 to 200 days, 0.63 kg/day from 200 to 400 days, 0.71 kg/day from 400 to 600 days and 0.38 kg/day from 600 to 800 days. Differences in live weight between the lightest and heaviest sires was 19 kg at 131 days (range 110.6–129.7 kg, inter-quartile range 115.7–121.0 kg) and increased to 90 kg at 800 days (range 454.5–544.8 kg, inter-quartile range 492.7–514.8 kg, Figure 2).

### 3.3. Comparison of Progeny Means for Live Weight at Different Ages

Least-squares means of the progeny groups were compared at different ages, and the scatterplots and estimates of the regression coefficients are shown in Table 3 and Figure 3.

## 4. Discussion

The aim of this experiment was to evaluate the post-weaning growth of Angus and Hereford sires via progeny testing of beef-cross-dairy offspring born to dairy cows and grown on hill country pasture.

Beef-cross-dairy cattle in the present experiment entered the beef system at 131 days, slightly older and heavier than the typical 84–100 days old and 100 kg live weight weaner calves from dairy breeds purchased by beef finishers [19,20]. At 200 days, progeny mean live weight in this study did not reach the minimum live weight reported by a recent Beef Progeny Test in New Zealand at 200 days, which ranged from 199.9 to 212.3 kg for beef cattle [21]. Cattle in this study were removed from their dams within 24 h of birth, group fed on an allowance of 4 L of milk/head/day and weaned at a minimum of 85 kg live weight [16]. This resulted in weaning occurring at a mean age of 82 days (range 47–119 days, or approximately 1.5–4 months old), compared to the weaning age reported in the Beef Progeny Test of 200 days (or approximately 6–7 months old). Furthermore, young cattle (recently weaned) require pasture with greater than 11.4 MJME/kgDM and covers higher than 2200 kgDM/ha to grow at 1 kg/day [22], or alternatively, they need to be supplemented with concentrate feed to increase liveweight gain [23]. Calves in this study were on low-quality pasture during the summer season (averages of 9.2 MJME/kgDM, CP 12.6% DM and NDF 55.8% DM). Therefore, lower growth rates to 200 days of beef-cross-dairy calves compared with beef-breed calves were expected, given the reduced time they were fed milk combined with the low-quality feed on offer.

Higher growth rates followed after 200 days. It has been previously shown that the loss in performance during scarcity of feed (or a dry season) could be proportionately compensated for in the following wet season when feed becomes abundant [24]. At 400 days, beef-cross-dairy cattle had weights in the range of 264.4–314.9, which were heavier than those reported in the Beef Progeny Test of 261.0–288.2 kg [21]. At 600 days, progeny mean live weights were between 401.2 and 464.5 kg with a mean of 427.0 kg, which were lower than live weights for sires in the Beef Progeny Test in New Zealand at 18 months of age (approximately 550 days) with a mean of 437.4 kg and a range of 416.8–455.2 kg [21]. Lastly at 800 days, least-squares means of the progeny groups were in the range of 454.5–544.8 kg, with a mean of 503.6 kg live weight. Assuming a typical value of around 50% dressing-out [22], the 503.6 kg live weight would yield a 251.8 kg carcass weight, or considering the lightest and heaviest sire in this study, they would yield a 227.3 to 272.4 kg carcass (45 kg difference). These weights are consistent with the range of carcass weights for New Zealand in 2019: an average of 242 kg/head for heifers and 313 kg/head for steers [1]. This indicates that there are beef sires that can be used on dairy cows to produce cattle of comparable growth to cattle from the beef industry.

Growth trajectories differed among beef sires used to produce beef-cross-dairy cattle on grazing hill country conditions. Differences between the lightest and heaviest sires increased from entry to the beef farm (131 days in this study) to finishing (800 days). This was expected, as differences in mature size drive different growth rates from early ages. Calves entered the beef system recently weaned and at similar live weights, because weaning occurred at a minimum of 85 kg live weight [16], which meant that lighter or slower growing calves were fed milk for more days to achieve the target weaning weight, and so may have had an advantage compared with the heavier or faster growing calves. As time progressed, each animal’s genetic capability to grow was expressed, and consequently, differences in liveweight amongst sires increased. Cattle from smaller-framed sires would have reached physiological maturity at an earlier age and at a lighter weight, while cattle from large-framed sires would have been less mature at comparable ages, thus leaner, growing faster and with greater final weights [25,26].

Cattle growth rate in this study was similar to New Zealand industry values of around 0.6–0.7 kg/day [22]. However, the rate of daily growth in this study was likely an underestimation of the growth potential of the cattle, because favourable environmental conditions are necessary for the full expression of the individual’s genetic capacity [27]. The low-quality pasture in the present experiment, particularly in the summer months (ME 9.2 MJ/kgDM, CP 12.6% DM, NDF 55.8% DM), is likely to have restricted animal performance. This has been shown across different sheep and beef farms in New Zealand, but particularly in regions where summer temperature is high [28]. Animals were unable to consume enough feed to meet nutrient requirements of energy and protein for high growth, due to the slow digestion of fibre and low protein content of mature pasture [29]. Therefore, the 90 kg difference in live weight between the lightest and heaviest sires at 800 days could have been greater if pasture or diet with higher quality was offered.

Even though growth of calves was likely restricted prior to 200 days, live weight at 200 days explained 51–56% of the variation in live weight at 400 and 600 days. This is because weights at different ages are highly correlated and related to adult size [26]. Farmers could use the 200 days live weight to estimate mature weights and therefore, time and feed required to achieve good finishing conditions to send the animal to slaughter. On the contrary, the live weight at 131 days explained little of the total variation of live weight later in life. It is likely that the live weight at 131 days reflected the artificial rearing and set weaning weight implemented rather than differences in genetic growth.

One limitation of this experiment was the scarce information on the dams, and so maternal breed was not accounted for. Dams were predominantly Holstein-Friesian or Holstein-Friesian-Jersey crossbred. A greater proportion of Holstein–Friesian genetics would produce heavier calves with faster growth rates, while a greater proportion of Jersey genetics would produce smaller animals with slower growth rates [30,31,32]. Nevertheless, it is unlikely that there would be a bias favouring particular sires in the data from this study because sires were rotationally allocated to mating days and randomly allocated to cows in oestrus on each mating day, and cows had similar live weights, body condition scores and milk production regardless of the sire they were bred with [16,33].

## 5. Conclusions

Live weight of beef-cross-dairy cattle on grazing hill country conditions differed among sires. There are beef sires that could be used on dairy cows to produce cattle of comparable growth to cattle from the beef industry. Even though growth of beef-cross-dairy calves was likely restricted to 200 days, live weight at 200 days explained a large proportion of live weights at 400 and 600 days. Differences between the lightest and heaviest sires increased from entry to the beef farm to finishing, and this difference at 800 days may have been greater if pasture of higher quality was available. Thus, the use of genetically superior beef sires has the potential to increase the growth of cattle born in dairy farms for meat production.

## Figures and Tables

**Figure 1 animals-10-02313-f001:**
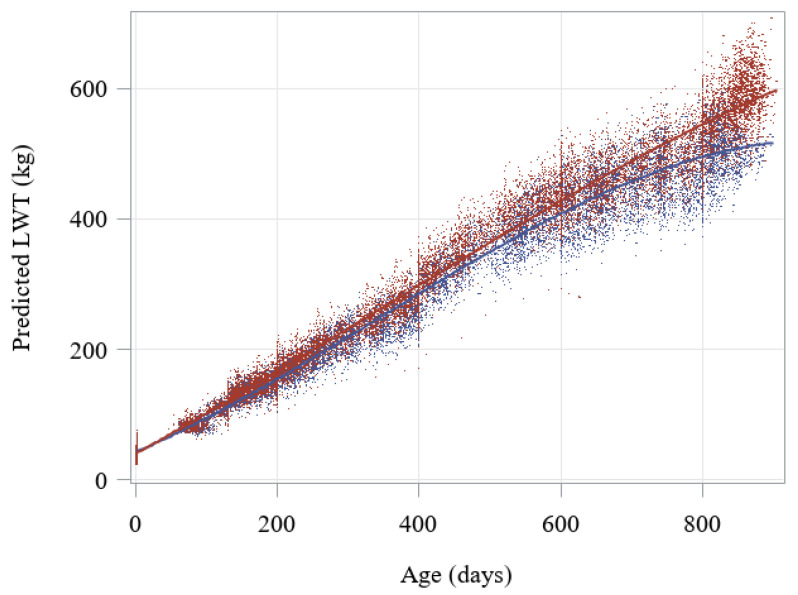
Predicted live weight (LWT, in kilograms) across age for beef-cross-dairy cattle (*n* = 1101 with a total of 33,111 predicted weights calculated with centred moving average of 3 values), according to sex (▪ heifers in blue, ▪ steers in red).

**Figure 2 animals-10-02313-f002:**
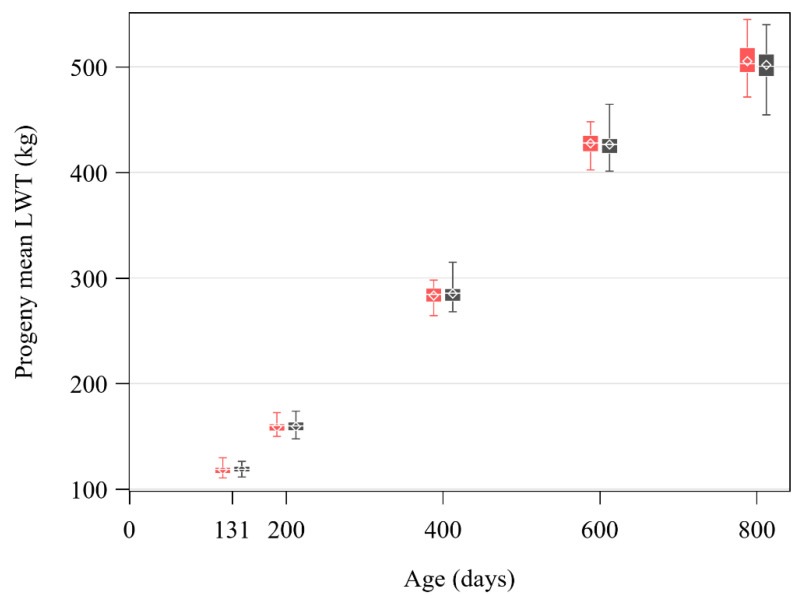
Distribution of least-squares means of the progeny groups of sires (Angus, ■ grey; Hereford, ■; red) for live weight (LWT) at 131, 200, 400, 600 and 800 days of age. Each box represents the inter-quartile range (25th to 75th percentiles), with the median value indicated by a line and the mean value indicated by a marker (◇). Whiskers extend to the minimum and maximum values.

**Figure 3 animals-10-02313-f003:**
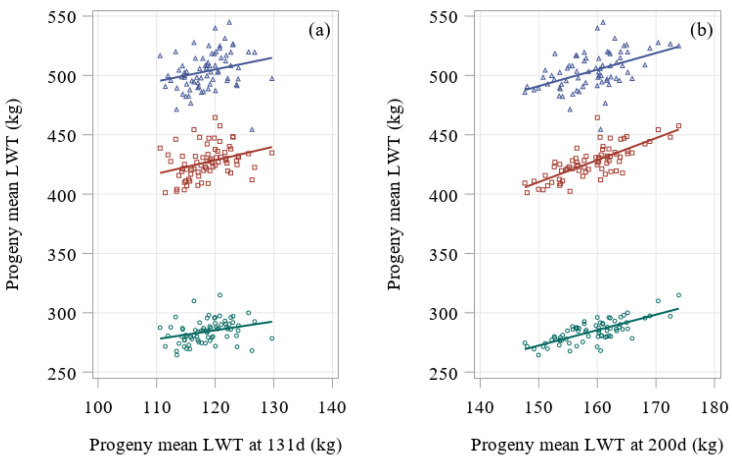
Regression of least-squares means of the progeny groups for live weight (LWT, in kilograms) at 400, 600 and 800 days of age, on least-squares means of the progeny groups for live weight at either (**a**) 131 or (**b**) 200 days of age. Individual sires are represented by one data point, at 400 (○ green circles), 600 (□ red squares) and 800 (∆ blue triangles) days of age. The respective regressions are indicated by solid lines.

**Table 1 animals-10-02313-t001:** Estimated breeding values (EBVs; mean ± SD) for live weights, at 200, 400 and 600 days of age, for 37 Angus and 40 Hereford sires [17].

Trait	Angus			Hereford		
	*n*	EBV	EBV Range	*n*	EBV	EBV Range
200d weight (kg)	37	39 ± 10	(18 to 59)	40	30 ± 7	(18 to 48)
400d weight (kg)	37	75 ± 14	(41 to 110)	40	54 ± 13	(31 to 79)
600d weight (kg)	37	96 ± 21	(49 to 135)	40	73 ± 19	(35 to 114)

*n*: number of sires used at mating; final number of sires included for data analysis: 34 Angus and 39 Hereford.

**Table 2 animals-10-02313-t002:** Number of progeny, mean (±SD) of the least-squares means of the progeny groups for live weight (LWT, in kilograms) and *p*-value for the effect of sire on live weight at 131, 200, 400, 600 and 800 days of age, for 73 sires.

Age (Days)	*n*	Progeny Mean LWT (kg)	Sire Effect *p*-Value
131	1101	118.6 ± 3.9	0.002
200	1086	159.1 ± 5.5	<0.001
400	1069	284.2 ± 9.6	<0.001
600	1035	427.0 ± 13.5	<0.001
800	913	503.6 ± 15.9	<0.001

*n*: number of progeny for all sires included.

**Table 3 animals-10-02313-t003:** Estimates of regression coefficients (intercept and slope) of least-squares means of the progeny groups for live weight (LWT, in kilograms) at 400, 600 and 800 days of age, on least-squares means of the progeny groups for live weight at 131 or 200 days of age, for 73 sires.

Progeny Mean	LWT 131 Days				LWT 200 Days			
	Intercept	Slope	*p*-Value	*R* ^2^	Intercept	Slope	*p*-Value	*R* ^2^
LWT 400d (kg)	236 ± 33	0.4 ± 0.3	0.149	0.03	82 ± 23	1.3 ± 0.2	<0.001	0.51
LWT 600d (kg)	333 ± 44	0.8 ± 0.4	0.035	0.06	144 ± 30	1.8 ± 0.2	<0.001	0.55
LWT 800d (kg)	461 ± 57	0.4 ± 0.5	0.466	0.01	296 ± 53	1.3 ± 0.3	<0.001	0.18

*R*^2^: coefficient of determination.

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
