# Peer review of "Sire Effects on Post-Weaning Growth of Beef-Cross-Dairy Cattle: A Case Study in New Zealand"

_animals, 2020, doi:10.3390/ani10122313_

Round 1
Reviewer 1 Report
In many European countries as well as in New Zealand, permanent grasslands account from 10% to almost 70% of the total agricultural areas. They are a powerful reservoir of feed for ruminants, provided they are properly cared for. For this reason, the presented article has great cognitive importance. In dairy herds, farmers provide milk production by mating the cows with the best dairy sires, use crossbreeding with a dual-purpose breeds in order to improve the health traits of the herd, or inseminate the cows with sires of beef breeds to improve the meat yield of the offspring. Presented publication focuses to describe this third way of increasing beef production in dairy herds. The authors performed analysis of the fattening performance of both heifers and bullock crossbreeds with Angus and Hereford sires. A statistical analysis was carried out to estimate the finishing body weight of animals at the age of 400, 600 and 800 days on the basis of the body weight at 131 days and 200 days of age. The obtained results are consistent with the results described by other authors. The added value of this work is the analysis of the fattening ability of animals on mountain pastures, the efficiency of which was strongly dependent on weather conditions. Currently, the method of beef production is being explored without the use of concentrates in fattening, for which cattle compete with humans. In the presented work, I only miss the analysis of the calving ease, as most of the beef breeds used for crossing with HF cows have a negative impact on this trait.
I would also suggest to move description of the "calf meal" from line 141 to 121.
At what replacement rate would you suggest using beef sires for mating in a dairy herd? What is more important there milk or meat?
Author Response
Point 1: In the presented work, I only miss the analysis of the calving ease, as most of the beef breeds used for crossing with HF cows have a negative impact on this trait.
Response 1: Calving ease was analysed as part of a previous publication from this study [1]. The following sentence was added in the Materials and Methods section (line 123): “Fewer than 1% of calves born to mixed-aged cows and fewer than 7% of calves born to 2-year-old cows were assisted at calving [16]”.
Point 2: I would also suggest to move description of the "calf meal" from line 141 to 121.
Response 2: The calf meal described in line 141 (now line 145) was supplemented post-weaning while the animals were grazing on summer-dry hill country pasture. A description of the calf meal supplemented pre-weaning has been added in line 125: “(CP 17% to 20% [16])”.
Point 3: At what replacement rate would you suggest using beef sires for mating in a dairy herd? What is more important there milk or meat?
Response 3: It depends. Typically in NZ, 26% of calves are kept as replacement dairy heifers [2,3]. With increasing use of sexed semen, fewer cows will need to be bred to produce replacement heifers, and so beef-cross-dairy production could grow. The priority of a dairy farmer is the health and survival of the dairy cow and their milk production, so meat, at the moment, is a by-product of secondary importance [2]. An evaluation of the most appropriate dairy herd mating system for generating beef-cross-dairy calves was outside the scope of this study, so no changes have been made on the manuscript.
- Coleman, L.W. The use of high genetic merit Angus and Hereford bulls in a New Zealand dairy herd. Thesis presented in partial fulfilment of the requirements for the degree of Doctor of Philosophy in Animal Science, Massey University, Palmerston North, New Zealand, 2020.
- Cook, A. The hunt for the missing billion: NZ's dairy beef opportunity. In Kellogg Rural Leaders Programme, Univesity, L., Ed. Lincoln, New Zealand, 2014.
- Hickson, R.E.; Zhang, I.L.; McNaughton, L.R. BRIEF COMMUNICATION: Birth weight of calves born to dairy cows in New Zealand. In Proceedings of Proceedings of the New Zealand Society of Animal Production, Dunedin, 2015; pp. 257-259.
Reviewer 2 Report
I would like to thank the authors of this manuscript for doing a good job in their presentation style, it was difficult to find an error.
The focus of this study is of high interest to the beef industry and brings a connection between the dairy and beef industries. It is of particular interest to learn that beef sires can produce progenies within the dairy structure that competes in performance with similar progeny in the beef structure by taking advantage of heterosis. However, much of the assessment is based on the sire side.
The authors noted that they were unable to assess the effect of the dairy dams on the progeny performance due to lack of information on the dam side. However, I believe that an understanding of the dam's effect on the progeny performance is critical for implementation of this idea. One way to navigate this will be to assess the effect of breed proportions of the various dams used on the growth trajectory of their progenies. This will provide a little insight into the effect of the underlying dairy/dam genetics on the progeny performance.
Thanks.
Author Response
Point 1: The authors noted that they were unable to assess the effect of the dairy dams on the progeny performance due to lack of information on the dam side. However, I believe that an understanding of the dam's effect on the progeny performance is critical for implementation of this idea. One way to navigate this will be to assess the effect of breed proportions of the various dams used on the growth trajectory of their progenies. This will provide a little insight into the effect of the underlying dairy/dam genetics on the progeny performance.
Response 1: The authors agree with this statement. Understanding the dam effect on the progeny growth is important. However, it is usual in the dairy industry to allocate lower merit or poorly recorded dams to beef bulls, with the best dairy cows retained for generating replacement dairy heifers. Therefore, at the current time, many of the beef-sired calves produced are from cows for which little is known about them. The study was conducted under commercial conditions, and unfortunately, dam records were inaccurate (breed proportions, parentage, breeding worth and breeding values). Previously published results from this study [1,2] did include dam phenotype traits (bodyweight, milk yield, body condition score) recorded during the experiment, and these analyses showed no difference regardless of the sire the dams were bred to. In addition, the random allocation of sires to mating days and cows in oestrus on each mating day, would have restricted any bias distribution. Consequently, although we couldn’t account for individual dam effects, it was expected that the dam effects would have been even across sires and breeds of sire. This was stated as a limitation of the study in lines 290-297 (now lines 296-303), and therefore no changes have been made in the manuscript. Further work using better recorded cows is planned to address these aspects.
- Coleman, L.W.; Back, P.J.; Blair, H.T.; Lopez-Villalobos, N.; Hickson, R.E. Milk production and rebreeding performance of mixed-aged dairy cows mated to Angus or Hereford bulls. New Zealand J. Anim. Sci. and Prod. 2019, 79, 144-148.
- Coleman, L.W. The use of high genetic merit Angus and Hereford bulls in a New Zealand dairy herd. Thesis presented in partial fulfilment of the requirements for the degree of Doctor of Philosophy in Animal Science, Massey University, Palmerston North, New Zealand, 2020.
Reviewer 3 Report
The experiment addressed in the paper aimed to evaluate the performance of Angus and Hereford sires via progeny testing of beef-cross-dairy offspring born to dairy cows and raised on New Zealand hill country pasture. Although this subject is not innovative, it is interesting and relevant since it portrays a concrete situation, concerning the chosen breeds and the conditions where the experiment was conducted. For this reason, I suggest mentioning some reference to this in the title - eg. "Sire effects on post-weaning growth of beef-cross-dairy cattle - a case study in New Zealand".
It may not be an original topic but it is important for animal production, especially in a country where dairy production plays an important role, such as New Zealand. Since the calf is a by-product of milk production, studies that show different performance and efficiency according to the paternal lines used will be an asset for production.
The article is written in a clear and concise manner, addressing the variables of interest for the study itself in an objective and easy to read way. The objectives are clear and the results allow conclusions to be drawn about them.
Author Response
Point 1: Although this subject is not innovative, it is interesting and relevant since it portrays a concrete situation, concerning the chosen breeds and the conditions where the experiment was conducted. For this reason, I suggest mentioning some reference to this in the title - eg. "Sire effects on post-weaning growth of beef-cross-dairy cattle - a case study in New Zealand".
Response 1: The manuscript title has been changed according to the suggestion, to: “Sire effects on post-weaning growth of beef-cross-dairy cattle: a case study in New Zealand”.